# Stability of Bituminous Emulsion Induced by Waste Based Bio-Surfactant

**Michele Porto, Paolino Caputo \*, Abraham A. Abe, Valeria Loise \* and Cesare Oliviero Rossi**

Department of Chemistry and Chemical Technologies, University of Calabria, Via P. Bucci, Cubo 14/D, 87036 Rende, Italy; michele.porto@unical.it (M.P.); abraham.abe@unical.it (A.A.A.); cesare.oliviero@unical.it (C.O.R.)

\* Correspondence: paolino.caputo@unical.it (P.C.); valeria.loise@unical.it (V.L.)

**Abstract:** In the asphalt industry, bituminous emulsions are widely used in road pavement operations and in building/construction processes such as cold mix asphalt and waterproofing processes, respectively. A very important fact to keep in mind is that not all types of bitumen are suitable for the realization of bituminous emulsions. This is largely due to the variation in their chemical nature and the different cracking processes carried out on the bitumen during the fractional distillation process in the petroleum industry. The objective of this study is to identify the underlying causes of the non-emulsionability of bitumen using Nuclear Magnetic Resonance (NMR) and Dynamic Shear Rheology (DSR) analysis. NMR analysis aims at identifying the fundamental chemical components that are responsible for the emulsionability of the bitumen binder and how important their role is in this phenomenon. On the other hand, the DSR analysis is aimed at determining if the rheological (viscoelastic) behavior of bitumen is implicated in its emulsionability. The indications gotten from the data produced by these techniques, enable us as soon as the analyzed bitumen is deemed non-emulsionable to identify what type of additive can be used to modify the bitumen and alleviate its non-emulsionability until a point where its chemical components become ideal for the realization of bituminous emulsions. In this research work, a model bitumen (labelled as Cimar) which is known for its excellently high emulsionability in the production of anionic bituminous emulsions was used as the reference sample. Two bitumens (labelled as Adriatica and Alma) which from preliminary testing were deemed non-emulsionable were alongside the additives selected and subjected to the aforementioned techniques for analysis on their emulsionability. The NMR data obtained allowed the identification of the chemical nature of the components of the analyzed bitumens and the design of the right additive which improves the bitumen and makes it suitable for the preparation of emulsions. In addition to these, a largely uncommon however effective method of acid number determination of bitumen gave indications on an underlying factor which largely influences the emulsionability of bitumen. An aliphatic and an aromatic surfactant were identified thanks to the spectroscopic findings in this study.

**Keywords:** bituminous emulsion; rheological properties; nuclear magnetic resonance (NMR) spectroscopy

## 1. Introduction

Bitumen is a viscoelastic material derived from the petroleum industry and for this reason its chemical composition is hugely dependent on the initial crude oil and the cracking process carried out on it. It is not miscible with water and this feature is fundamental for the production of emulsions. From a chemico–physical point of view, the emulsification process involves the dispersion of a fluid in another provided that the two are not miscible with each other. When this phenomenon occurs, one of the two fluids breaks into drops, while the other exists as a continuous medium. Generally, emulsions are of two

types: water in oil or oil in water, depending on whether the dispersing part (continuous medium) is water or oil [1,2]. Emulsions of this type are thermodynamically unstable systems; in fact, their free energy decreases over time [3]. However, it is possible to stabilize an emulsion with the use of emulsifiers [4,5].

The latter are surfactants, i.e., molecules with a polar (hydrophilic) head and a nonpolar (lipophilic) tail. Thanks to their amphiphilic nature, surfactants are adsorbed at the interface of the two phases, arranging themselves with the polar head in the aqueous phase, and with the non-polar tail in the organic phase. This mechanism facilitates the reduction of the interfacial tension between the droplets and the dispersing phase [6]. Bituminous emulsion is a worldwide technology, and its first application was patented in the early 1900s [7] being mainly used for pavement maintenance and road repair. Bituminous emulsions are oil-in-water emulsions in which droplets of bitumen are dispersed in the continuous aqueous phase. In order to reduce the interfacial tension and prevent coalescence phenomena, ionic surfactants are used. The salt form of the fine surfactant is solubilized in water before its addition to the bitumen. The two fluids are poured simultaneously into a colloidal mill, whose rotation speed is between 1000 and 6000 rpm. This speed causes the breaking of the bitumen into small droplets, typically between 1 and 10 μm in diameter, readily covered by the emulsifier. [8–10]. The main uses of bitumen emulsions are in road building, roofing and waterproofing [11–14].

Bituminous emulsions are commonly studied in terms of stability, asphalt performance, breaking times [15–17]. Emulsions can be identified according to their chemical nature, basicity or acidity.

The innovative endpoint of the research work behind this article is to make bitumens emulsifiable or rather to understand why some bitumens are not emulsifiable. In this work, only basic emulsions are investigated, without compromising the generality of the proposed methodological approach. Not all types of bitumen used in the road paving sector are emulsifiable under basic conditions. This work therefore aims to identify the chemico–physical causes of the non-emulsifiability of the aforementioned bitumens and to find suitable additives capable of making them emulsifiable. [14].

## 2. Materials and Methods

### 2.1. Chemicals and Materials

Three pristine bitumens from different sources are tested in this work: one with a penetration grade of 70/100 from Venezuela (labelled as Cimar), used as the reference sample, the second one with a penetration grade of 70/100 from an Italian refinery—Alma Petroli S.p.A. (labelled as Alma) and the third one also from an Italian refinery—Adriatica Bitumi S.p.A. (labelled as Adriatica). All three bitumens were supplied by Cimar Produzione S.r.l. (Italy); more information is reported in Table 1. A commercial wax (SASOBIT) and a waste from animal fat processing processes supplied by S2A Soluzioni Ambientali in the form of liquid pitches which was labelled as LP were used as additives. Another additive, an aliphatic/aromatic acid surfactant labelled as AS which is used in the field of cosmetics and soap production, supplied by Kimical S.r.l. (Italy) was also used. All of these aforementioned substances were used as additives in this study.

**Table 1.** Physical properties of bitumens used in this study.

| Measured Properties | Standard | Unit | Cimar 70/100 | Adriatica 170/210 | Alma 70/100 |
|---|---|---|---|---|---|
| Penetration at 25 °C | EN 1426 | 0.1 mm | 66 ± 1 | 185 ± 1 | 68 ± 1 |
| Softening point | EN 1427 | °C | 47.8 ± 0.2 | 42.6 ± 0.2 | 47.2 ± 0.2 |
| Flash point | EN 2592 | °C | ≥230 | ≥220 | ≥230 |
| Solubility | EN 12592 | % (m/m) | ≥99 | ≥99 | ≥99 |

### 2.2. Sample Preparations

In general, the emulsions prepared are composed of 50–60% by weight of bitumen and 40–50% of water.

The percentage of emulsifier (1–6%) is weighed based on the total weight of the emulsion. During the emulsion process, the pH was kept stable at 14 thus all the prepared emulsions are basic. This emulsifier was supplied by Cimar Produzione S.r.l. (Italy).

Asphaltene Determination

In a vial, a specific mass of bitumen was dissolved in an equal volume in milliliters of chloroform ($CHCl_3$), for example, 3 g of bitumen in 3 mL of $CHCl_3$. Thereafter, n-pentane which was forty times the $CHCl_3$ volume was added to the solution (in the case of 3 mL of $CHCl_3$, 120 mL of n-pentane would be added). The solution was left in a dark chamber for two hours with occasional homogenization of the solution. The asphaltenes which were precipitated were then filtered using a filter paper (Whatman 42 ashless) in a funnel under vacuum conditions. The residue gathered on the filter paper was then washed several times with n-pentane until the solvent became clear and colorless, thus evidencing the absence of any other component other than the asphaltenes. Finally, the filter paper was dried in an oven at 80 °C for three hours and then the residue of the solvent was removed in vacuum for two hours. Just for further facultative analysis, the filtrate containing the maltene fraction was then evaporated to dryness with a rotary evaporator under reduced pressure and the residual solvent was removed under a vacuum pump [18]. This maltene sample was stored for further analysis.

### 2.3. Rheological Characterization

Dynamic Shear Rheological (DSR) measurements on bitumen samples were carried out using a controlled shear stress rheometer (SR5, Rheometric Scientific, Piscataway, NJ, USA) equipped with a parallel plate geometry (gap 2 mm, $\varphi = 25$ mm within the temperature range 25–150 °C) and a Peltier system (±0.1 °C) for temperature control [19].

### 2.4. NMR Measurement

The $^1$H-NMR spectra were recorded at room temperature on a high-resolution Bruker Avance 500 MHz spectrometer (11.74 T) (Bruker, Rheinstetten, Germany) equipped with a 5-mm TBO probe (Triple Resonance Broadband Observe) and a standard variable-temperature unit BVT-3000. The $^1$H-NMR experiments were performed on bitumen diluted in Carbon Tetrachloride ($CCl_4$), in order to avoid overlapping with a possible proton signal of the solvent. All three analyzed bitumens showed four different peaks referable to protons as reported by Oliviero Rossi et al. [18].

### 2.5. Bitumen Acidic Number Determination Method

This is a thermometric catalytic titration method carried out according to ASTM D8045 standards. It was used to determine the end point of a chemical reaction (in this case between bitumen in solution and KOH) through the use of a temperature measuring probe and the addition of a chemical to enhance the detection of the endpoint.

## 3. Results and Discussion

Before examining the chemical composition of the bitumen, we calculated the acid number of each sample correlating the values obtained to the emulsionability of each bitumen. A relatively simple method known as Catalytic Thermometric Titration (as described in Section 2.5) was used to determine the acid number of the bitumen samples used for this study. The main advantage of this method is that it is relatively easy to carry out and this method can be adopted by industries who need to determine if a certain bitumen to be used for the production of emulsions is up to emulsionable standards.

As mentioned earlier, this procedure can be carried out in any laboratory and by any technician who understands the basic principle of titration. As per instrumentation, only basic titration apparatus and a temperature measuring probe are needed. In Table 2, we show the results obtained.

**Table 2.** Acid number for each sample.

| Bitumen | Acid Number (mg/g KOH) |
|---|---|
| Cimar | 22.7 |
| Alma | 13.5 |
| Adriatica | 14.7 |

As expected, making the emulsion as indicated in Section 2.2, we observed that the Cimar bitumen is the only one among the three analyzed bitumen samples to be emulsionable. The high acidity of Cimar bitumen (the only emulsionable one) helps us to understand how this parameter could govern the process of bituminous emulsion formation. Hence, the acidity can be used as a key role to evaluate the tendency of a generic bitumen sample to produce a stable emulsion. We proceeded with the characterization of all samples with the aim of identifying the ideal additives which will make the non-emulsionable bitumen samples emulsionable.

Time cure curves (Figure 1) acquired for the three neat bitumens show that, as expected, the two bitumen samples with the same 70/100 penetration grade have very similar rheological behaviors, while Adriatica bitumen has a lower sol–gel transition temperature of about 10 °C.

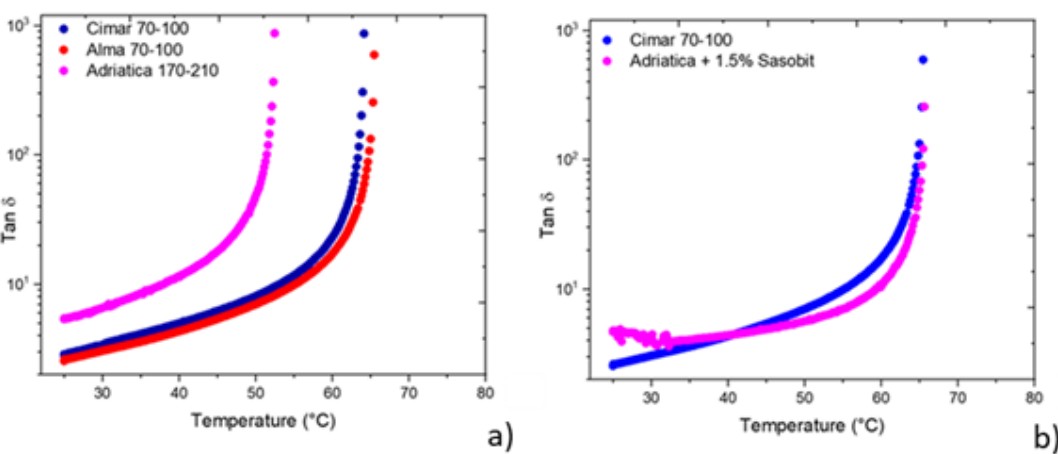

**Figure 1.** (**a**) Time cure test for Cimar 70–100 (blue); Alma 70–100 (red) and Adriatica 170–210. (**b**) Comparison between the time cure test for Cimar and modified Adriatica with 1.5% SASOBIT.

In Table 3, the transition temperature and percentage asphaltene of all the bitumen samples are reported.

The asphaltene content could play a role in the stability of emulsions; in fact, asphaltenes show an important interfacial activity [20].

**Table 3.** Transition temperature and percentage asphaltene of all bitumen samples.

| Bitumen | Transition Temperature ± 0.1 (°C) | Asphaltene ± 1 (%) |
|---|---|---|
| Cimar | 64.1 | 17 |
| Alma | 64.4 | 31 |
| Adriatica | 52.4 | 19 |

Adriatica bitumen shows a different rheological behavior but similar asphaltene concentration to the reference bitumen. Therefore, in order to obtain a mechanical behavior similar to that of the reference bitumen, only the Adriatica bitumen was treated with several modifier additives. On the market, there are several types of modifying additives with only some of them chemically interacting with the bitumen [21–25]; others simply impart their own chemico–physical properties to it [26–33].

The criteria on which the additive was chosen was its ability to increase the transition temperature without increasing the viscosity of the binder at high temperatures [34–36].

This commercial additive is a wax-based additive called SASOBIT. In fact, at temperatures above 100 °C it acts as a fluxing agent, lowering the viscosity of the binder thus favoring the emulsion process. This additive was added to the Adriatica bitumen in different dosages. In Figure 1b, we reported only the time cure of the best sample obtained with 1.5% additive. This modified bitumen was treated according to the reference experimental conditions for emulsion production. Nevertheless, even in this case, no stable emulsion was obtained proving that the mechanical behavior does not play a decisive role in the formation of bituminous emulsions.

Therefore, having established that the mechanical properties do not significantly affect the emulsifiability of the bitumen, a new approach based on the chemical composition of the bitumen was undertaken. Still using Cimar bitumen as reference, NMR analyses were carried out to obtain more information on the chemical composition of all three bitumens analyzed [18,37–39].

The acquisition of the proton NMR spectra (Figure 2) allowed the characterization of the bitumen under examination from a chemical point of view. Table 4 shows the fractional proton distribution obtained from the normalization and integration of the peaks.

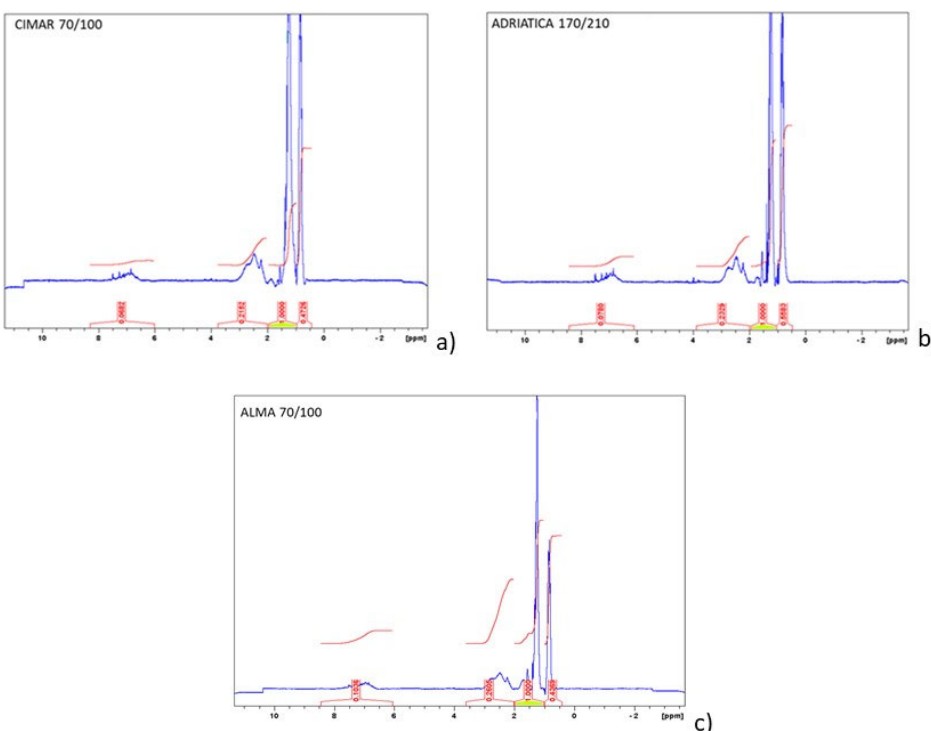

**Figure 2.** $^1$H-NMR spectra of Cimar bitumen (**a**), Adriatica bitumen (**b**) and Alma bitumen (**c**).

**Table 4.** Fractional proton distribution of Cimar, Adriatica and Alma bitumens.

| Sample | Hydrogen Distribution ± 0.05 | | | |
|---|---|---|---|---|
| | $H_{ar}$ | $H_{\alpha}$ | $H_{\beta}$ | $H_{\gamma}$ |
| **Cimar Bitumen** | 3.88 | 12.26 | 56.95 | 26.91 |
| **Adriatica Bitumen** | 4.17 | 12.46 | 53.50 | 29.87 |
| **Alma Bitumen** | 5.75 | 14.46 | 55.52 | 24.26 |

By comparing the distribution of the protons of the bitumens analyzed, it emerges that the Cimar bitumen has a similar aromatic character (3.88%) to the Adriatica one (4.17%) while the Alma bitumen has greater aromatic character (5.75) in comparison with the other bitumens. Furthermore, Cimar bitumen is characterized: (i) by having the highest percentage of hydrogens in $\beta$ compared to the aromatic ring (56.95% against 55.52% and 53.50%); (ii) having a percentage of hydrogens in $\alpha$ (12.26%) in the aromatic ring comparable to that of Adriatica bitumen (12.46%) but lower than that of Alma bitumen (14.46%); (iii) finally, the percentage of methyl hydrogens is intermediate between the percentages of the other two bitumens (26.91% against 24.26% and 29.87%). The major differences correspond to methylene hydrogens in $\beta$ to the aromatic ring and methyl protons in the $\gamma$ position. From the information obtained by NMR spectroscopy, it was possible to identify any surfactants that could compensate for the chemical differences between the bitumens.

Adriatica bitumen has a different rheological profile from the reference bitumen, however it has a similar value of percentage asphaltenes and shows values of aromatic components very close to it. This data pointed towards the choice of an additive capable of making the bitumen more acidic without altering the aromatic components, which is an acid aliphatic/aromatic surfactant. In light of this, the commercial additive called "AS" was chosen. This additive is a Brønsted acid surfactant [40,41], consisting of an acid group (polar head), as a substituent of an aromatic ring, and a long hydrophobic chain (lipophilic tail) [42,43]. Thanks to these characteristics, the AS additive has been able to balance both the aromatic and aliphatic components to compensate for the differences with Cimar bitumen.

In fact, once the Adriatica bitumen was modified with AS at different dosages (2.5 to 7.5%), it was possible to proceed with the preparation of the emulsion in accordance with what was done for the emulsion obtained with the Cimar bitumen. This modified bitumen was able to produce stable emulsions and the best emulsion was obtained after modification with 2.5% AS (see Figure 3).

Alma bitumen has a rheological profile very similar to Cimar bitumen (Figure 1a) but has a different percentage of asphaltenes (almost double, see Table 2) [44]. Above all, based on the data obtained by NMR measurements, it is observed that it has a higher aromatic component. Consequently, the function of the additive to be used to make this bitumen more similar to the reference bitumen must be to integrate the maltene part, thus decreasing the asphaltene percentage, lowering the percentage of the aromatic components and also increasing the total acidity of the bitumen with additives. The latter is in fact a central component in the realization of anionic emulsions.

The methodological approach used shows how fundamentally important the chemical nature of bitumen is for the realization of bituminous emulsions. The NMR data obtained give us guidance on the choice of the coadjutant potential of the emulsifier. This step also showed us the importance of determining the acidity of bitumen.

For all these reasons, an aliphatic acidifier called "LP" was identified as a suitable additive. This additive is ecofriendly and facilitates the development and promotion of a circular economy. Its nature is mainly aliphatic and gives an acidic character to the bitumen system and is mainly made up of a mixture of fatty acids (surfactants) [45,46].

The Alma bitumen was modified with LP additive at different dosages (4 to 10%) before proceeding with the preparation of the emulsion in accordance with what was

done. This modified bitumen was able to produce stable emulsions and the best emulsion was obtained after modification with 4% LP (see Figure 4).

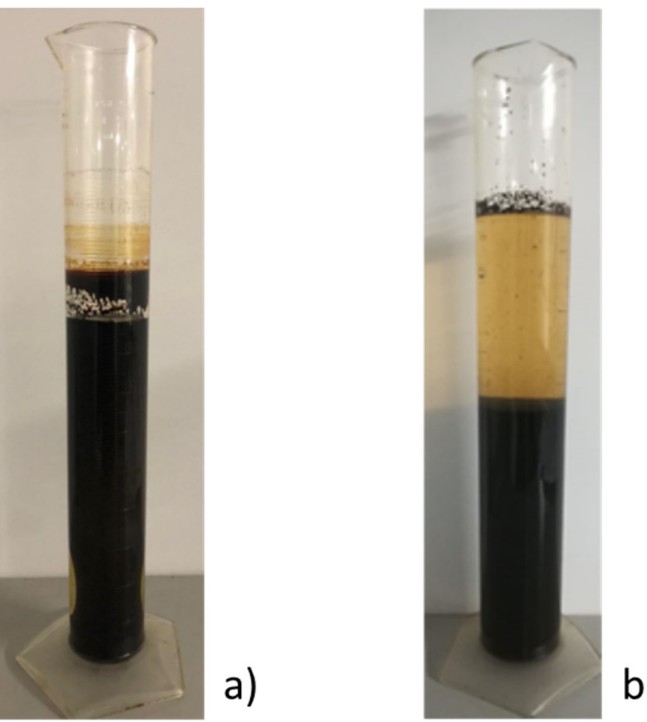

**Figure 3.** Pictorial representation of Adriatica bitumen emulsions obtained with aliphatic/aromatic acid surfactant (AS) additive (**a**), and without AS additive (**b**).

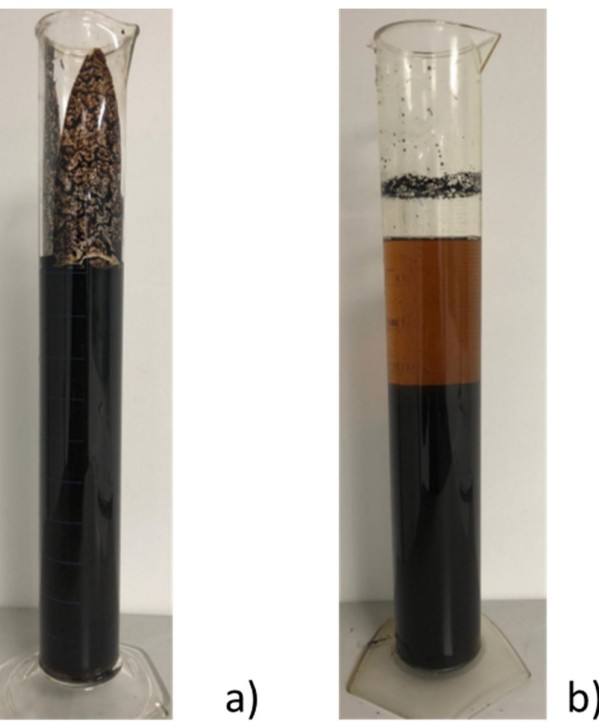

**Figure 4.** Pictorial representation of Alma bitumen emulsions obtained with liquid pitches (LP) additive (**a**), and without LP additive (**b**).

## 4. Conclusions

There are emulsifiable and non-emulsifiable bitumens on the market and the aim of this study was to identify the type of additive suitable for making non-emulsifiable bitumens emulsifiable.

Two types of approaches were used, one involving the study of the mechanical characteristics and the other a study of the chemical composition of the binders.

The first approach was achieved through rheological analysis and led to the conclusion that the mechanical properties of bitumen do not depict its emulsifiability. In fact, Alma bitumen has a rheological profile similar to that of the reference bitumen but is not emulsifiable. On the other hand, the Adriatica bitumen initially had a different rheological profile from the reference bitumen and was subsequently made similar to the reference through the use of suitable additives. Nevertheless, the Adriatica bitumen remains non-emulsifiable.

The second approach, thanks to the data obtained with NMR measurements, allowed the identification of the problems of the two non-emulsifiable bitumens.

As mentioned earlier, the acid number determination is a useful tool for researchers and most especially for industry personnel to know the acid number of bitumen which they intend to use for production processes. At the moment, we are trying to identify the acid number value limit under which a bitumen sample is non-emulsionable.

In conclusion, we can affirm that, thanks to an approach based on the study of the chemical composition of the various bitumens, it is possible to identify any deficiencies present in the sample and identify additives capable of bridging these gaps which make a non-emulsifiable bitumen suitable for the realization of anionic emulsions.

In the case of the bitumen analyzed in this work, the use of the appropriate additives, LP for Alma bitumen and AS for Adriatica bitumen, made it possible to make both bitumens emulsifiable. Both these additives are eco-friendly and, in the future, have the potential to substitute harmful substances that are used in industrial processes.

**Author Contributions:** Formal analysis (M.P.), investigation, data curation, writing-original draft (P.C.), formal analysis, writing-review & editing (A.A.A.), writing-review & editing (V.L supervision, writing-original draft, writing-review & editing (C.R). All authors have read and agreed to the published version of the manuscript.

**Funding:** This research received no external funding.

**Institutional Review Board Statement:** Not applicable.

**Informed Consent Statement:** Not applicable.

**Data Availability Statement:** The data presented in this study are available on request from the corresponding authors.

**Acknowledgments:** Thanks to Cimar Produzione S.r.l. company (Italy) both for the supply of raw materials and for the contribution and support given for the preparation of the emulsions.

**Conflicts of Interest:** The authors declare no conflict of interest.

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
