# Peer review of "Stability of Bituminous Emulsion Induced by Waste Based Bio-Surfactant"

_applsci, doi:10.3390/app11073280_

Round 1

Reviewer 1 Report

Journal: Applied Sciences

Title: Stability of bituminous emulsion induced by new bio-surfactant: Physical chemistry characterization.

This paper investigated the potential application of two emulsifiers on improving the storage stability property of unstable emulsified bitumen, which is important for pavement engineering researchers to focus and further work regarding the use of stable emulsified bitumen. However, this manuscript should be revised carefully before published. Here are some suggestions for helping further improving this paper.

Comments:

1.The abstract section should be rewritten.

2. In introduction part, the research gap between your and previous studies need to be added here, rather than introducing the definition and preparation method of emulsified bitumen.

3. The physical properties of three bitumen should be supplemented here, including penetration, softening point, ductility and viscosity. In addition, the chemical properties or representative molecular structure of the two chemical emulsifiers could be shown in your paper.

4. Regarding the asphaltene determination, please describe the detailed extraction method here. And why do you need to measure the asphaltene dosage in three bitumen? How about the influence of asphaltene content on the storage stability of corresponding emulsified bitumen?

5. Why is the NMR test needed in your research?

6. The acid number method of bitumen should be removed to characterization methods part.

7. The English editing should be enhanced. It’s better to find a native researcher to help you improve it.

8. The references in your paper is good. However, it would be better to         further add some relevant literatures. Here are some literatures provided        to help you further understand the preparation of modified and        emulsified bitumen. 

a. Rheological properties, compatibility, and storage stability of SBS latex-modified asphalt. Materials. 2019, 12(22), 3683.

b. Utilization of wax residue as compatibilizer for asphalt with ground tire rubber/recycled polyethylene blends. Construction and Building Materials. 2020. 230, 116966.

c. On the rejuvenator dosage optimization for aged SBS modified bitumen. Construction and Building Materials. 2021, 271, 121913.

9. Some result should be further discussed.

Author Response

WE USE THIS MEDIUM TO THANK THE REFEREES FOR THEIR SUGGESTIONS WHICH HAVE MADE IT POSSIBLE FOR US TO IMPROVE THE QUALITY OF THE MANUSCRIPT.

Referee 1

This paper investigated the potential application of two emulsifiers on improving the storage stability property of unstable emulsified bitumen, which is important for pavement engineering researchers to focus and further work regarding the use of stable emulsified bitumen. However, this manuscript should be revised carefully before published. Here are some suggestions for helping further improving this paper.

Comments:

1.The abstract section should be rewritten.

The abstract has been rewritten in a more detailed and comprehensive manner.

  1. In introduction part, the research gap between your and previous studies need to be added here, rather than introducing the definition and preparation method of emulsified bitumen.

We agree with the referee and the introduction part has been extended following his indication

  1. The physical properties of three bitumen should be supplemented here, including penetration, softening point, ductility and viscosity. In addition, the chemical properties or representative molecular structure of the two chemical emulsifiers could be shown in your paper.

We added more information about the bitumen used in the study (see tab.1).

  1. Regarding the asphaltene determination, please describe the detailed extraction method here. And why do you need to measure the asphaltene dosage in three bitumen? How about the influence of asphaltene content on the storage stability of corresponding emulsified bitumen?

We agree with the referee and we described the method used to determine the asphaltene content and we emphasized why the asphaltene content is important for emulsions according to the literature data

  1. Why is the NMR test needed in your research?

We thank the referee for his question. We needed the NMR test to quantify the type of Hydrogens present (and consequently the nature of molecules) in all the three bitumens. This allows us to evaluate the most ideal additive to improve the bitumen’s emulsionability.

  1. The acid number method of bitumen should be removed to characterization methods part.

According to the referee we moved the Acidic number method to the Materials and Methods part.

  1. The English editing should be enhanced. It’s better to find a native researcher to help you improve it.

The manuscript has been checked by a native English-speaking researcher

  1. The references in your paper is good. However, it would be better to further add some relevant literatures. Here are some literatures provided to help you further understand the preparation of modified and emulsified bitumen. 
  2. Shisong Ren ,Xueyan Liu ,Weiyu Fan ,Haopeng Wang andSandra Erkens Rheological properties, compatibility, and storage stability of SBS latex-modified asphalt. Materials. 2019, 12(22), 3683.
  3. Utilization of wax residue as compatibilizer for asphalt with ground tire rubber/recycled polyethylene blends. Construction and Building Materials. 2020. 230, 116966.
  4. On the rejuvenator dosage optimization for aged SBS modified bitumen. Construction and Building Materials. 2021, 271, 121913.

We added the references suggested by the referee

  1. Some result should be further discussed.

We tried to improve the discussion

Reviewer 2 Report

Comments attached. 

Author Response

WE USE THIS MEDIUM TO THANK THE REFEREES FOR THEIR SUGGESTIONS WHICH HAVE MADE IT POSSIBLE FOR US TO IMPROVE THE QUALITY OF THE MANUSCRIPT.

Referee 2

SPECIFIC COMMENTS

  1. The authors should consider giving the name of the surfactant instead of calling it “New”

We changed the title in text.

  1. Would consider doing away with the “physical chemistry characterization” part of the title since it describes the methodology.

This has been done according to the referee’s specification.

  1. Page 1. “….application was patented in the early 1900s, mainly used for…” cite the source of this information.

We added the reference in the text

  1. Page 3. Give the full names of the chemical CCl4

This was also done

  1. Page 3. What standard did you use to determine that “we observed that the Cimar bitumen is the only one among the three analysed bitumen samples to be emulsionable”?

We thank the referee for the suggestion and we specify in the text that, as a standard, we used a bitumen given to us by the Cimar S.p.a. ( bitumen 70/100 from Venezuela) which is actually used in bitumen emulsion production. In particular, we have reproduced the emulsion with this bitumen to confirm the emulsionability in basic condition. Then we did the same for the other bitumens to evaluate their emulsionability in the same conditions.

  1. Write the names of AS and LP in full. What do the letters denote?

We better explained the acronym for AS and LP in the text.

  1. The report reads like a routine laboratory exercise. What new sphere of knowledge have the authors advanced?

This is a good observation. However, the goal of this study is to help the companies that use bitumen to produce basic emulsions to have the possibility to choose a wide range of bitumens not limiting themselves to a specific kind of bitumen. In particular, we tried to find the best additive for each bitumen tested in order to enhance their emulsionability.

  1. The science behind the study has not come out clearly.

We are sorry for that, we tried to stress how the scientific approach suggested by us can be useful to investigate the bituminous emulsions. Physical chemistry techniques are powerful tools to identify additives able to make the bitumen emulsionable   

Round 2

Reviewer 2 Report

  1. The abstract presents no results, conclusion or recommendations.
  2. Make reference to Table 1 in the text.
  3. Figure 1 comes several paragraph after being referred to. Bring it closer to where it is referred.
  4. Your discussion of results is dominated by literature sources. Ideally, you should be giving your perspective of the scientific meaning of the results

Author Response

Referee 2 round 2

  • The abstract presents no results, conclusion or recommendations.

We thank the referee for the suggestions. We added text which highlight results and conclusions gotten from the study.

  • Make reference to Table 1 in the text.

This was done according to the referee’s suggestion.

  • Figure 1 comes several paragraph after being referred to. Bring it closer to where it is referred.

This was done according to the referee’s suggestion.

  • Your discussion of results is dominated by literature sources. Ideally, you should be giving your perspective of the scientific meaning of the results

We changed the text.

  • English language and style are fine/minor spell check required

The manuscript was revised by Mother tongue.
